# Label-Free Electrochemical Aptasensor for the Detection of the 3-O-C_12_-HSL Quorum-Sensing Molecule in *Pseudomonas aeruginosa*

**DOI:** 10.3390/bios12070440

**Published:** 2022-06-22

**Authors:** Denisa Capatina, Teodora Lupoi, Bogdan Feier, Adrian Blidar, Oana Hosu, Mihaela Tertis, Diana Olah, Cecilia Cristea, Radu Oprean

**Affiliations:** 1Department of Analytical Chemistry, Faculty of Pharmacy, “Iuliu Hatieganu” University of Medicine and Pharmacy, 4 Pasteur Street, 400349 Cluj-Napoca, Romania; denisa.elen.capatina@elearn.umfcluj.ro (D.C.); teodora.lupoi@elearn.umfcluj.ro (T.L.); blidar.adrian@umfcluj.ro (A.B.); hosu.oana@umfcluj.ro (O.H.); mihaela.tertis@umfcluj.ro (M.T.); roprean@umfcluj.ro (R.O.); 2Department of Infectious Diseases and Preventive Medicine, Faculty of Veterinary Medicine, University of Agricultural Sciences and Veterinary Medicine Cluj-Napoca, Calea Manastur 3-5, 400372 Cluj-Napoca, Romania; diana.olah@usamvcluj.ro

**Keywords:** quorum sensing, *P. aeruginosa*, aptamer, electrochemical aptasensor, electrochemical detection, 3-O-C_12_-HSL

## Abstract

*Pseudomonas aeruginosa*, an opportunistic Gram-negative bacterium, is one of the main sources of infections in healthcare environments, making its detection very important. N-3-oxo-dodecanoyl L-homoserine lactone (3-O-C_12_-HSL) is a characteristic molecule of quorum sensing—a form of cell-to-cell communication between bacteria—in *P. aeruginosa*. Its detection can allow the determination of the bacterial population. In this study, the development of the first electrochemical aptasensor for the detection of 3-O-C_12_-HSL is reported. A carbon-based screen-printed electrode modified with gold nanoparticles proved to be the best platform for the aptasensor. Each step in the fabrication of the aptasensor (i.e., gold nanoparticles’ deposition, aptamer immobilization, incubation with the analyte) was optimized and characterized using cyclic voltammetry, differential pulse voltammetry, and electrochemical impedance spectroscopy. Different redox probes in solution were evaluated, the best results being obtained in the presence of [Fe(CN)_6_]^4−^/[Fe(CN)_6_]^3−^. The binding affinity of 106.7 nM for the immobilized thiol-terminated aptamer was determined using surface plasmon resonance. The quantification of 3-O-C_12_-HSL was performed by using the electrochemical signal of the redox probe before and after incubation with the analyte. The aptasensor exhibited a logarithmic range from 0.5 to 30 µM, with a limit of detection of 145 ng mL^−1^ (0.5 µM). The aptasensor was successfully applied for the analysis of real samples (e.g., spiked urine samples, spiked microbiological growth media, and microbiological cultures).

## 1. Introduction

*Pseudomonas aeruginosa (P. aeruginosa)* represents one of the most opportunistic and common Gram-negative bacteria, reported as the main source of infections in humans in various healthcare environments (e.g., nursing homes, hospitals). *P. aeruginosa* is the causative agent for a wide range of infections, such as cystic-fibrosis-associated lung disease; endocarditis; meningitis; urinary tract, post-surgical, ocular, ear, skin, soft tissue, or chronic wound infections; and community-acquired infections (usually in immunosuppressed hosts). This bacterium has a high ability to develop intrinsic and acquired antimicrobial resistance, and to form biofilms, reducing the treatment options and leading to increased rates of severe infections and mortality. All of these factors make the rapid detection of *P. aeruginosa* of utmost importance [1,2].

Quorum sensing (QS) is a form of cell-to-cell communication between bacteria, which allows individual cells to gather information about the bacterial cell population density and the environmental conditions, enabling the switch to a collective behavior as a community and the adaptation to changing conditions in the environment. The communication between bacteria occurs through autoinducers (AIs)—small signaling molecules produced by bacteria and found in the extracellular medium in amounts proportional to the population density [3]. *P. aeruginosa* produces the following AIs: N-3-oxo-dodecanoyl L-homoserine lactone (3-O-C_12_-HSL), N-butyryl L-homoserine lactone (C_4_-HSL), 2-heptyl 3-hydroxy-4-quinolone (PQS), and 2-(2-hydroxyphenyl) thiazole-4-carbaldehyde (IQS). The bacteria can detect the AIs’ concentration and, when a threshold concentration of these QS molecules is reached, they modify their gene expression, causing a change in their behavior, such as the activation of certain metabolisms, secretion of virulence factors, or biofilm formation. Therefore, the detection of QS molecules can be a useful tool for monitoring the bacterial population and its pathogenicity [3,4,5].

As extracellular small molecules, AIs are found in the nM range in biological fluids such as sputum, urine, and plasma [6,7,8], and in the low μM range in biofilms and culture media [9,10,11], requiring very specific and sensitive methods for their detection and quantification in these complex matrices containing various ions, many types of proteins, and carbohydrates [4].

High-performance liquid chromatography coupled with mass spectrometry (HPLC–MS) or tandem mass spectrometry (HPLC–MS/MS) is the conventional approach used for the separation and detection of QS molecules [4,7,11,12,13], but other promising approaches—including enzyme-linked immunosorbent assay (ELISA) [10,14,15] and genetically engineered whole-cell bacterial biosensors [16,17,18]—have been introduced in the last decade to identify these molecules.

Even though these methods present good analytical performance, they display some disadvantages; they require expensive laboratory instruments and reagents, long analysis time, extensive sample handling, complex working protocols, and the use of expensive and polluting solvents, and cannot be used to perform decentralized analyses [4,12]. Fortunately, the electrochemical (bio)sensors can overcome these limitations, due to their many advantages (e.g., available cost, rapid analysis, biocompatibility, high sensitivity, and selectivity) [19,20].

Moreover, screen-printed electrodes (SPEs) offer the possibility of miniaturization and in situ analyses [21]. Furthermore, gold nanoparticles (AuNPs) are very popular for the fabrication of electrochemical platforms, due to their properties such as good conductivity, large surface area, biocompatibility, and ease of modification and functionalization [22].

Several electrochemical approaches have been described for the detection of PQS and its precursor 2-heptyl-4-quinolone (HHQ) [23,24,25], but few for the detection of acyl homoserine lactones (AHLs), such as 3-O-C_12_-HSL and C_4_-HSL [26].

In recent years, biomimetic systems, such as molecularly imprinted polymers (MIPs) and aptamers (APTs), have attracted considerable interest due to their affinity for target molecules, and are increasingly used in the development of biosensors. *Jiang* et al. [27] described an electrochemical sensor based on magnetic MIPs for the detection of AHLs in bacterial supernatant samples. The methods showed high reproducibility, specificity, and sensitivity, with a limit of detection in the nM range.

Another promising direction in the development of (bio)sensors for QS detection is the use of APTs as biorecognition elements. The APTs are short single-stranded DNA or RNA oligonucleotides, artificially selected for their capacity to bind with high specificity to a target molecule [19]. APTs present some superior characteristics compared to antibodies, such as high stability in extreme conditions (e.g., temperature, pH, medium composition), easy synthesis and functionalization, and retaining the ability of binding behavior after immobilization. These properties make APTs suitable options for the elaboration of sensors [19].

The combination of electrochemical techniques with APTs has promoted the development of aptasensors with many applications in electrochemical sensing for a wide range of molecules (e.g., drugs, biomolecules, toxins, pesticides), ions, and pathogens [4,19,20,28]. The electrochemical aptasensing device usually consists of an APT-based sensing layer, a transducer that converts the resulting chemical signal into an electrical output, and an analyzer that has a readout function [28]. Depending on the electrochemical techniques used and the signal measured (impedance, current, or potential), aptasensors can be impedimetric, amperometric/voltammetric, or potentiometric [19,20]. Various electrochemical aptasensors have been developed based on different measurement techniques, such as differential pulse voltammetry (DPV), cyclic voltammetry (CV), electrochemical impedance spectroscopy (EIS), and field-effect transistors (FET). Electrochemical aptasensors can be divided into labeled (with enzymes, antibodies, metal nanoparticles, or redox compounds) or label-free aptasensors, depending on the detection methods used. The labels can be covalently or non-covalently attached to APTs. In both cases, the presence of targets is indicated by changes in either current or capacitance, and the concentration of the analyte can be directly correlated with the change in the electrochemical feature [20,28]. Label-free aptasensors are favored, to the detriment of label-based sensors, because of their advantages—such as less complicated design, reduced preparation time, analytical quality, wider dynamic range, relatively fast reaction time, easy operation, and cost-effectiveness by eliminating the need for complex labels [29,30]. It is important to note that label-free electrochemical aptasensors, in addition to their noticeable advantages, have some disadvantages, such as lower selectivity than similar labeled electrochemical aptasensors, and a low signal-to-noise ratio at low target concentrations, requiring mediator composites to amplify the output signal [29].

Overall, electrochemical aptasensors have several advantages, such as high sensitivity and specificity, low cost, ease of use, and the possibility of miniaturization, making them useful tools for biomedical, food, or environmental analysis in the field. However, one challenge in developing an aptasensor for a particular target is the availability of the specific APT for that analyte [19]. An APT with a good capacity to specifically bind the 3-O-C_12_-HSL molecule has recently been described, and it has been used to block the QS in *P. aeruginosa* [31].

In the present work, the development of a new electrochemical label-free aptasensor for the detection of 3-O-C_12_-HSL, using a carbon-based SPE modified with AuNPs by electrodeposition, is described. Each stage in the development of the aptasensor (i.e., AuNP deposition, APT immobilization, and the incubation process) was optimized. Each step in the development of the aptasensor was characterized using CV, DPV, and EIS, and evaluated using different redox probes in solution, such as 1,1′-ferrocenedimethanol (FcDM), hexaammineruthenium chloride (III) (Ru(NH_3_)_6_Cl_3_), methylene blue (MB). and the ferrocyanide–ferricyanide couple ([Fe(CN)_6_]^3−/4−^). The binding affinity for the immobilized APT was determined using surface plasmon resonance (SPR) analyses. The optimal configuration of the aptasensor exhibited a logarithmic range from 0.5 to 30 µM, with a limit of detection (LOD) of 145 ng mL^−1^ (0.5 µM). The aptasensor was successfully used for the analysis of real samples (e.g., spiked urine samples, spiked microbiological growth media, and microbiological cultures). To the best of our knowledge, this is the first report of an APT-based sensor for the detection of the 3-O-C_12_-HSL QS molecule.

## 2. Materials and Methods

### 2.1. Materials

All chemicals used in this study were of analytical grade, and were used without further purification. HCl, NaOH, NaCl, KCl, MgCl_2_, and 6-mercapto-1-hexanol (MCH) were purchased from Merck (Branchburg, NJ, USA); H_2_SO_4_, potassium ferrocyanide K_4_[Fe(CN)_6_], potassium ferricyanide K_3_[Fe(CN)_6_], Tris(hydroxymethyl)aminomethane (TRIS), MB, Ru(NH_3_)_6_Cl_3_, FcDM, 3-O-C_12_-HSL, C_4_-HSL, N-dodecanoyl-L-homoserine lactone (C_12_-HSL), N-(3-Oxodecanoyl)-L-homoserine lactone (3-O-C_10-_HSL), and acetaminophen (APAP) were purchased from Sigma-Aldrich (Louis, MO, USA); gentamicin sulfate (GEN) was purchased from BioWorld (Louis Park, MN, USA); ceftazidime pentahydrate (CFZ) was received from Antibiotice SA Iasi (Iasi, Romania). A label-free single-stranded thiol-tethered DNA APT was used for specific recognition of the QS molecule with the following sequence of nucleotides: 5′-GCA-ATG-GTA-CGG-TAC-TTC-CCG-GGG-CCC-GCT-TCT-GGT-GCG-GTG-TAC-TAG-TGA-CCG-CAA-AAG-TGC-ACG-CTA-CTT-TGC-TAA-(CH_2_)_6_-SH-3′, synthesized by Eurogentec (Liege, Belgium). All solutions were prepared with UltraPure DNase/RNase-Free distilled water (Thermo-Fisher, Waltham, MA, USA).

The APT was received as a lyophilized powder and was dissolved in TRIS buffer, pH 7.4, containing 10.0 mM TRIS, 100.0 mM NaCl, 100.0 mM KCl, and 10.0 mM MgCl_2_. The stock solution was then divided into aliquots and stored at −20 °C. Before the experiments, the APT aliquots were diluted to the desired concentration with TRIS buffer, and underwent thermal shock by heating at 96 °C for 5 min, followed by cooling down to −20 °C for 1 min. The selected APT showed good affinity and specificity for 3-O-C_12_-HSL, with a previously reported dissociation constant (*K_D_*) of 20–35 nM [31]. The supporting electrolytes in this study were 0.1 M KCl and TRIS buffer.

For the analysis of the real samples, urine was collected from a healthy volunteer, and the nutrient broth (NB) used for the spiked samples was Cooked Meat Medium (Oxoid, Hampshire, UK). Two strains of *P. aeruginosa*—a standard reference strain (ATCC 27853) and a clinical isolate—were cultivated for the analysis on culture media. Each of the strains was inoculated in 12 mL glass tubes containing 10 mL of NB. Cultures were subjected to aerobic incubation for 72 h at 37 °C.

### 2.2. Instruments

The electrochemical experiments were performed using an Autolab PGSTAT 302N (Metrohm Autolab, Utrecht, The Netherlands) equipped with the associated Nova 1.10.4 software developed by Methrom (Utrecht, The Netherlands), and a PGSTAT-12 potentiostat (Eco Chemie Autolab) equipped with GPES 4.9 software. The SPEs used had carbon (C-SPE) and gold (Au-SPE) as working electrodes; both types had a silver pseudo-reference and a carbon-based counter electrode, and were purchased from Metrohm-DropSens (Oviedo, Spain).

All SPR experiments were performed on a three-channel Biosensing BI-2500 instrument (Biosensing Instrument Inc. (Tempe, AZ, USA)). To extract the SPR kinetic parameters, the kinetic data were analyzed in the framework of the Langmuir isotherm/1:1 binding model using BI-software version 2.4.4 (Biosensing Instrument Inc., Tempe, AZ, USA), which includes the software Scrubber (BioLogic Software Pty Ltd., Campbell, Australia).

### 2.3. Methods

#### 2.3.1. Aptasensor Development

The first step in obtaining the aptasensor was the AuNPs’ electrodeposition. A solution of 1.5 mM HAuCl_4_ prepared in 0.5 M H_2_SO_4_ was used for the electrodeposition by CV, where the potential was cycled 35 times between −0.2 V and 1.2 V, with a scan rate of 0.1 V s^−1^. This method was adapted from the work of Tertis et al. [32].

Once the C-SPE/AuNPs surface was obtained, the APT immobilization was performed with a 1 μM APT solution. We tested two immobilization techniques: one involved leaving the APT solution overnight on the electrode surface, and the other used multipulse amperometry (MPA), applying two steps of potential (−0.2 V and +0.5 V for 45 ms each, with a total duration of 300 s). After this step was achieved, the surface of the electrode was always covered with a drop of TRIS buffer to avoid dehydration and spatial degeneration of the APT. The blocking step was carried out with a 100 μM MCH solution prepared with TRIS buffer through a similar electrochemical process as the APT multipulse-assisted immobilization, but the total duration was shortened to 50 s.

Incubation with the target molecule took place on the benchtop, at room temperature, in a hermetic box to avoid evaporation, for a duration of 30 min. After each modification step, the electrode was washed three times with 50 μL of nuclease-free water. To avoid non-specific absorption, after the incubation with 3-O-C_12_-HSL, the electrode was washed five times with 200 μL of nuclease-free water.

#### 2.3.2. Electrochemical Methods

Several electrochemical techniques were applied for the characterization of each step. The DPV method used a start potential of −0.3 V and an end potential of +0.6 V for the anodic scan, and vice versa for the cathodic scan, with a scan rate (SR) of 0.01 V s^−1^, pulse height (PH) of 0.1 V, and pulse width (PW) of 25 ms. The EIS was measured in a frequency range (FR) from 0.01 to 100,000 Hz for a total of 50 frequencies, at open-circuit potential values. The CV parameters were as follows: −0.5 to +0.6 V potential window, SR of 0.05 V s^−1^, and a step potential (SP) of 0.00244 V. The indirect characterizations utilized 5 mM [Fe(CN)_6_]^3−/4−^ prepared in a 0.1 M KCl aqueous solution as the redox probe.

#### 2.3.3. Analysis Procedure/Quantification Method

The electrode surface was characterized after each modification step by EIS and DPV. After the blocking step with MCH, the electrode was washed three times with 50 μL of TRIS buffer. An EIS analysis was performed to establish the resistance at the surface of the electrode after the blocking step (**R_ct MCH_**), using the [Fe(CN)_6_]^3−/4−^ couple as the redox probe. The EIS was followed by a DPV analysis, and the current intensity (**I_MCH_**) was measured. After the two analyses, the electrode was washed again and prepared for incubation.

A stock solution of the target molecule was first prepared in ethanol, and then the desired concentrations were obtained by diluting the stock solution with nuclease-free water. A volume of 50 μL from the corresponding concentration of the target molecule solution was applied to the three electrodes of the screen-printed support. The incubation took place at room temperature for 30 min, in a closed box, making sure that the electrodes were stable and secured from drought.

Once the incubation was over, the electrodes were washed five times with 200 μL of TRIS buffer to avoid unspecific adsorption of the molecule. The resistance (**R_ct HSL_**) and the current intensity (**I_HSL_**) were measured again by EIS and DPV. In EIS, the signal was calculated as the percentage decrease in resistance after the incubation, using Equation (1). For the DPV analysis, the signal was calculated as the percentage increase in current intensity after the incubation with 3-O-C_12_-HSL, using Equation (2):(1)$$S{\;}_{incubation}=\frac{\left(R_{ct\;MCH}-R_{ct\;HSL}\right)*100}{R_{ct\;MCH}}$$
(2)$$S{\;}_{incubation\;}=\frac{\left|I_{MCH}-I_{HSL}\right|*100}{I_{MCH}}$$

The molecule interacts with the APT, leading to a decrease in resistance. The more molecules there are to bind to the APT, the greater the electron transfer and the greater the difference between **R_ct MCH_** and **R_ct HSL_**, leading to a more intense signal. The same phenomenon happens in DPV where the current intensity increases, so the difference |**I_MCH_** − **I_HSL_**| is more significant when the concentration of the analyte is higher.

#### 2.3.4. SPR Analysis

The running buffer for the bonding experiments was 10 mM TRIS buffer, pH 7.4, and the flow rate was set to 60 μL/min. To extract the kinetic parameters, the kinetic data of the APT–analyte interactions were analyzed in the framework of the Langmuir isotherm/1:1 binding model, and also using a steady-state affinity model. For the kinetic parameters, the calculated association rate constant (*ka*) and dissociation rate constant (*kd*) were used to determine the kinetic dissociation constant (*K_D_*_,*kinetic*_) through the defining relationship *kd/ka*. An equilibrium dissociation constant (*K_D_*_,*equilibrium*_) was also obtained from the steady-state affinity model and compared with the *K_D_*_,*kinetic*_. The smaller the *K_D_*, the greater the affinity of the APT for its target [28,33].

The immobilization of the APT on the SPR sensing chip was conducted based on the Au–S interaction. The exposed Au surface of the bare Au chip provided by Biosensing Instrument Inc. (Tempe, AZ, USA) was covered with a solution of 1 μM thiolated APT and left overnight at 4 °C to form the APT self-assembled monolayer. Afterwards, the Au–APT surface was rinsed with TRIS buffer and incubated again with a solution of 100 μM MCH for 30 min at room temperature to block the unoccupied active binding sites. The modified chip was installed in the SPR instrument. After the stable baseline was achieved, different concentrations of 3-O-C_12_-HSL (10–1000 nM) were injected over the chip surface. For each cycle, 300 μL of the compound solution was injected at a constant flow rate (60 µL/min) and flowed through the surface, followed by a 300 s delay for dissociation (60 s), regeneration, and baseline stabilization. Before the first cycle, 300 μL of the running buffer was injected for blank subtraction.

#### 2.3.5. Interference Studies

The interference studies observed the influence of several active substances and other chemically related AHLs on the electrochemical signal of 3-O-C_12_-HSL. Firstly, the signal of a 10 μM 3-O-C_12_-HSL and each 10 μM interferent solution was calculated following the steps from Section 2.3.3. Thereafter, the signal of a mixture between 3-O-C_12_-HSL and APAP, CFZ, GEN, C_4_-HSL, C_12_-HSL, or 3-O-C_10_-HSL—both at a 10 μM concentration—was measured. The modification of the signal given by the mixture was calculated as the percentage of the signal given by the solution of 3-O-C_12_-HSL.

#### 2.3.6. Analysis of Real Samples 

For the analysis of biological samples, 3-O-C_12_-HSL was added to untreated urine samples to obtain a concentration of 20 μM. This solution was further diluted 1:1 with ethanol. A volume of 0.5 mL was collected and evaporated. The sediment was suspended in 1 mL of nuclease-free water, obtaining a solution with a concentration of 10 μM of the target molecule. This final solution was analyzed, and the concentration was calculated using the calibration curve. For the analysis of the culture medium, liquid NB was used. To 1 mL of NB, an adequate quantity of 3-O-C_12_-HSL was added to obtain a concentration of 25 μM. This solution was vortexed with 4 mL of ethanol; afterwards, 2 mL was sampled and underwent a process of evaporation. The sediment was suspended in 1 mL of nuclease-free water. The final solution that contained 10 μM 3-O-C_12_-HSL was then incubated on the aptasensor for 30 min.

For the analysis of *P. aeruginosa* cultures, a standard strain (ATCC 27853) and a clinical isolate were cultivated in 10 mL of NB. Cultures were submitted to aerobic incubation for 72 h, at 37 °C. According to the set protocol, the cultures were sampled, inactivated, and tested at 16 h, 24 h, 48 h, and 72 h. The electrochemical signal was measured using the same sample treatment as in the case of the spiked culture medium analysis. In parallel with the electrochemical analysis, the number of colonies was counted.

All procedures and analyses were performed following the principles of the Helsinki Declaration, and with the approval of the Ethics Committee of “Iuliu Hatieganu” University of Medicine and Pharmacy Cluj-Napoca, Romania (147/30.03.2020).

#### 2.3.7. Estimation of the Number of Microorganisms

At each testing point, a suspension was prepared by adding 4 mL of saline to 1 mL of initial bacterial suspension, with respect to the dilution attained by using 96% ethyl alcohol for inactivation. Decimal dilutions up to 10^−11^ were made due to the high turbidity of the suspension; 9 mL of saline was distributed in a series of sterile culture tubes; 1 mL of the bacterial suspension was added to the first tube, resulting in a 1/10 dilution. Using another pipette, 1 mL of the 1/10 dilution was passed into the proximate tube, resulting in 1/100 dilution, etc. Then, 1 mL of the solution corresponding to each dilution was poured into a Petri dish, and 12–15 cm^3^ of nutrient agar (Nutrient Agar, Tulip Diagnostics, Verna, India), melted and cooled to 45 °C, was subsequently added to the plates. Homogenization of the contents of the plates was achieved by rotational movements in the horizontal plane. The plates were then left for solidification for 30 min. Incubation was performed at 37 °C for 24 h. The colonies developed both on the surface and in the depths of the agar were counted following the incubation. We only evaluated the plates with a density of colonies that allowed counting and respected the dilution ratio. The number of bacteria that grew at 37 °C, at set time intervals, was expressed in colony-forming units (***CFU***), and was calculated using Equation (3):(3)$$TNG\;\left(CFU\;mL^{-1}\right)=\frac{\Sigma \left(n\;d\right)}{N}$$
where:
***n*** = the number of colonies developed on a Petri dish;***d*** = the dilution from which the respective plate was inoculated;***N*** = the number of Petri dishes considered; only plates in which the dilution ratio was observed were considered.

## 3. Results and Discussion

### 3.1. Elaboration and Optimization of the Aptasensor

The development of the aptasensor involved several steps: modification of the electrode surface with AuNPs, immobilization of the APT, blocking of the free sites with MCH, and interaction of the immobilized APT with the analyte. Figure 1 schematically shows the fabrication and the detection process of the proposed sensor.

The parameters that influence the aptasensor elaboration were studied, including the electrode material, the nanostructured platform, the APT immobilization, and the incubation of the molecule. The MCH blocking step was an adaptation of the APT immobilization, and did not undergo further optimization. A single parameter varied at any given time, and the best value was chosen by calculating the percentage of signal suppression or enhancement compared to the signal generated by the previous surface, before the modification step.

#### 3.1.1. Optimization of the AuNPs Platform

Two electrode materials were tested: an Au-SPE, and a C-SPE/AuNPs obtained by the electrodeposition of a HAuCl_4_ solution. Before the APT immobilization, the surface of the Au-SPE was activated by cycling the potential from −0.3 V to +1 V, 25 times, in 0.5 M H_2_SO_4_. In Figure 2, the electrochemical behavior of the two tested platforms can be followed. A descending trend was observed up to the blocking step with MCH, showing the successful immobilization of the APT and the deposition of MCH. After the incubation with 50 μM 3-O-C_12_-HSL, a decrease in the signal of only 13.28% was obtained, compared to the AuNPs generated from a HAuCl_4_ solution, which led to an increase of up to 68.32%, as seen in Table 1. For the C-SPE/AuNPs platform, the decrease in signal was more significant after the APT immobilization and MCH blocking, and the signal change after the incubation with 50 μM 3-O-C_12_-HSL was higher (Figure 2B,D).

To investigate the electrochemical features of the modified electrode after each step in the experimental protocol, and to confirm the voltammetry results, EIS was performed. The charge-transfer properties between the electrode surface and the electrolyte were evaluated in the presence of [Fe(CN)_6_]^−3/−4^ as a redox probe, and are presented as a Nyquist plot in Figure 2C for Au-SPE and in Figure 2D for C-SPE/AuNPs. This type of representation allows the evaluation of the charge-transfer resistance (**R_ct_**) between the redox probe in solution and the electrode surface, along with the diffusion-limiting process. The result shows that bare SPE presents a semicircle at high frequencies and a straight line at low frequencies (Figure 2C,D; black). There is an obvious decrease in the semicircle part observed after the activation of Au-SPE (Figure 2C), indicating that the surface is cleaner and more conductive, and has an effective surface area to facilitate electron transfer [34]. The diameter of the semicircle increased slightly after the immobilization of the APT (Figure 2C; blue), due to the partial coverage of a portion of the surface with APT molecules, causing an increase in the **R_ct_**. However, the subsequent blocking steps with MCH (Figure 2C; green) and immobilization of the target molecule (Figure 2C; red) did not cause a significant change in the appearance of the Nyquist diagrams or the **R_ct_** value.

In the case of the C-SPE/AuNPs platform, a significant decrease in **R_ct_** was obtained after modification of the carbon surface with AuNPs. These results indicate that the presence of AuNPs could enhance the effective surface area and conductivity. Thus, the kinetics of the electron transfer for the redox probe were facilitated at the modified electrode. The APT was subsequently self-assembled at the surface of the C-SPE/AuNPs. In this step, a dramatic increase in **R_ct_** was observed (Figure 2D; blue). This could have been due to prevention created for the redox probe by the negatively charged APT at the electrode [35]. A further increase in **R_ct_** was observed after the blocking step with MCH, which could be related to the covalent attachment of the molecule (Figure 2D; green). After the step involving the capture of the target analyte at the surface by specific interaction with APT, both the **R_ct_** value and the allure of the Nyquist diagram changed significantly, and a new semicircle appeared in addition to the original one, confirming both the change in the electron-transfer mechanism between the redox probe and the electrode surface, and the successful immobilization of the 3-O-C_12_-HSL molecules. This phenomenon is usually encountered when the electrochemical process involves more than one rate-determining step, and in this case, it may have been due to the presence on the electrode surface of the non-conductive molecules of 3-O-C_12_-HSL [36].

It can be observed that the EIS measurements confirmed the observations drawn from DPV; therefore, the C-SPE/AuNPs were chosen as the platform for the aptasensor.

For the generation of AuNPs, the concentration of HAuCl_4_ and the number of cycles for deposition were optimized. The best increase in the signal of the redox probe was observed for a concentration of 1.5 mM HAuCl_4_ and with 35 cycles.

#### 3.1.2. Optimization of the Aptamer Immobilization

The APT immobilization was based on the interaction between the thiol function attached at the 3′ end of the APT and the AuNPs deposited on the carbon surface of the electrode when a S–Au covalent bond is formed. The functionalization of AuNPs via simple thiolate chemistry is a widely used method in many research areas, and Au–S bond formation has been an intensively studied topic for decades due to its applications in sensor development [37]. The formation of the covalent Au–S bond is a complex process involving two stages: physisorption, with the initial formation of a coordinative bond between the -SH group and Au, while the H atom remains advantageously attached to the S atom; followed by chemisorption, involving the dissociation of the S–H bond and the formation of the covalent Au–S bond due to the deprotonation of thiols and the formation of thiol radicals. The dissociated hydrogen atoms can be adsorbed on the gold surface or released as H_2_. The pH of the medium can affect the formation of the Au–S bond. A neutral or alkaline medium favors the dissociation of the S–H bond and the formation of the covalent bond [38]. The technique and parameters of this modification step are important, as they influence the morphology of the APT layer and, therefore, the performance of the aptasensor. Two modification techniques have been used: overnight incubation, and MPA from a 1 μM APT solution. The potential-assisted method showed better results, with a signal decrease of 52.68% compared to the overnight incubation, resulting in a decrease in current intensity of 33.88% (Table 2). These results show that the MPA method promotes better immobilization of the APT and enhanced formation of Au–S bonds between the APT and AuNPs, leading to the assembly of a more organized and stable layer. The Au–S interaction and the efficiency of the immobilization method employed in this study were also reported by Ge et al. [39] and Jambrec et al. [40]. When shifting the potential of zero charge between two values, the DNA layer assembles rapidly and offers the possibility to obtain reproducible surfaces, which are essential for analytical performance.

Jambrec et al. [37] demonstrated that the kinetics of Au–S binding depend on the duration of the potential pulse and the thiol concentration. The total duration of the experiment was determined, and real-time chronoamperograms were recorded. It can be seen in Figure 3A that the stabilization of the current was reached after approximately 300 s, and that a time extension did not bring any improvement. The total duration of 300 s is also supported by the EIS analysis, when the greatest passivation of the electrode (433.26%) was obtained (Figure 3B), showing the high immobilization of the APT at the electrode surface.

The pulse duration was also varied, and it manifested an influence on the interaction of the APT with the AuNPs’ surface. A pulse duration of 45 ms was the optimal value, as it led to the most significant passivation of 43.06%, compared to 29.02% and 14.97% given by 30 ms and 60 ms, respectively (Figure 3D).

Another parameter optimized was the APT solution concentration, with three different concentrations of APT being tested: 1, 2, and 3 μM. The best bond between the surface and the APT was obtained using a concentration of 1 μM—a lower concentration leading to a better arrangement of the APT, allowing the immobilization of a greater amount of APT, reflected in the electrode passivation, as shown in Table 2 and Figure 3C.

#### 3.1.3. Optimization of the Incubation with 3-O-C_12_-HSL

The influence of the incubation time on the signal response was also investigated. The developed aptasensor was incubated in a static mode, by applying 50 μL of a standard solution containing a concentration of 10 μM 3-O-C_12_-HSL for different time intervals (Figure 4A). The difference in the signal before/after incubation increased with increasing incubation time in the first 30 min, but a slight decrease was observed thereafter—in both DPV and EIS. The results indicate that the APT on the electrode surface becomes saturated with the molecule after a certain time, and then the non-specific adsorption occurs. Therefore, an incubation time of 30 min was used in this work.

The efficiency of dynamic incubation was tested by introducing the aptasensor in a stirred solution of 10 μM 3-O-C_12_-HSL. A higher diffusion coefficient would favor the binding of more molecules of the analyte by the APT. In this particular case, the signal modification was more important when the incubation took place in static conditions, as shown in Table 2 and Figure 4B. One explanation for this is the possibility that the molecules already bound to the APT can be removed by stirring the solution.

In Table 2, the optimized parameters are summarized. All of the measurements were carried out in triplicate, and the values in Table 2 represent the average of the three measurements.

### 3.2. Aptasensor Characterization

#### 3.2.1. Electrochemical Characterization

To characterize the electrode surface after each modification step and obtain a good signal change after incubation with the target molecule, a suitable redox probe was chosen. Four redox probes with different charges were tested: neutral FcDM, positively charged Ru(NH_3_)_6_Cl_3_, negatively charged [Fe(CN)_6_]^3−/4−^, and MB, with high affinity for the nucleobases in the structure of the APT. It was expected that each modification of the electrode would influence the electron transfer between the working electrode and the redox probe. Each step was highlighted by all redox probes, but the best results were obtained for [Fe(CN)_6_]^3−/4−^.

In the case of FcDM, an increase in current intensity was observed after AuNP deposition, a decrease after APT immobilization, and a further decrease after MCH blocking; however, the increase in the signal after incubation with the analyte was insignificant (Figure 5A). For the second probe tested, Ru(NH_3_)_6_Cl_3_ manifested a repulsion towards the AuNPs layer, leading to a decrease in signal, and it failed to detect a difference between the APT immobilization and MCH blocking. The increase in signal given by the bound target molecules was not high, and the currents were small (Figure 5B).

After an incubation of 30 min with 0.5 mM MB of the C-SPE/AuNPs/APT/MCH, a significant signal was obtained (Figure 6; blue), showing the good affinity of the MB for the APT. The same incubation was adapted for a C-SPE/AuNPs/MCH platform, but no signal was obtained (Figure 6; green), proving again that the APT is responsible for binding the redox probe. The electrochemical signal was measured in 10 mM PBS pH 7.4 after multiple steps of washing to remove the free MB. After incubation with different concentrations of analyte, the MB-labeled aptasensor showed a decrease in signal with the increase in the concentration of the target molecule, as seen in Figure 6. The bonds between MB and the succession of nucleotides were not strong enough, and after successive incubations with the blank, the signal continued to decrease as the MB was removed; therefore, the initial decrease in the signal cannot be assigned to the analyte. The instability of these bonds makes the detection of 3-O-C_12_-HSL using MB as the redox probe impossible.

The electrode surface was electrochemically characterized in 5 mM [Fe(CN)_6_]^3−/4−^ after each modification step, using the following techniques: DPV in oxidation and reduction, CV, and EIS (Figure 7 and Figure 8). In DPV (oxidation and reduction) (Figure 7A,B, respectively), it can be seen that after the deposition of AuNPs, the peak current increased greatly, indicating that the modification of the electrode surface with AuNPs played an important role in increasing the electroactive surface area and the electrical conductivity, resulting in a significant improvement in electron transfer. Thus, it can be stated that the kinetics of the electron transfer between the electrode surface and the redox probe were improved by the deposition of AuNPs on the electrode surface. After the immobilization of the thiolated APT on the electrode surface, the peak current decreased significantly, indicating a blocking of the charge transfer at the electrode interface due to the non-electroactive properties of the APT. This behavior can be also attributed to the repulsion of the negatively charged redox probe by the negatively charged APTs, as reported previously in other studies [32,35,41]. The peak currents also decreased after the deposition of MCH, which was used to block the non-specific sites on the electrode surface. After the 30-minute incubation at room temperature with the analyte, an increase in the electrochemical signal in DPV was observed. This indicates a change in the conformation of the APT, leading to the facilitation of the electron transfer.

The corresponding CV curves in [Fe(CN)_6_]^3−/4−^ are shown in Figure 7C. The curves show that the redox probe [Fe(CN)_6_]^3−/4−^ exhibited a reversible CV at the C-SPE, with a peak-to-peak separation of about 188 mV (black). The functionalization of the electrode with AuNPs proved to be beneficial both as a result of the electrocatalytic effect correlated with the decrease in peak-to-peak separation at about 127 mV, but also by facilitating the electronic transfer, which determined the increase in the signal recorded for the redox probe with this surface (orange). Subsequent immobilization of the APT via the thiol–Au bond resulted in a decrease in the intensity of the oxidation/reduction signals, and the peak-to-peak separation increased to approximately 183 mV (blue). The same tendency to decrease the signal intensity was amplified after blocking the active sites unoccupied with APT remaining on the surface with MCH, the step after which peak-to-peak separation increased slightly, to approximately 239 mV (green). The last step of the experimental protocol is represented by the binding of the target analyte to APT through the specific interaction, which determines the proximity of the two peaks at about 132 mV and the increase in their intensity (red). This may be due to the change in the charge of the APT after binding the 3-O-C_12_-HSL molecules—a process that eliminated the steric hindrance for the diffusion of the redox probe to the electrode surface. Furthermore, the electrochemical behavior of the aptasensor was similar to that obtained in DPV: signal increase after deposition of AuNPs, signal decrease after immobilization of the APT and blocking of the electrode surface with MCH, and an increase in signal after incubation with the molecule of interest.

The results obtained via EIS were consistent with those obtained in DPV and CV. As can be seen in Figure 8, the resistance at the electrode surface decreased dramatically with the deposition of AuNPs, indicating the facilitation of the electron transfer. The APT immobilization resulted in a significant increase in resistance, indicating that the presence of the APT at the electrode surface strongly impairs the electron transfer. Moreover, the semicircle domain showed an additional increase after MCH deposition. After the incubation step, the resistance decreased significantly, showing a change in the conformation of the immobilized APT that specifically bound the analyte in the sample.

The EIS diagrams after each modification step in the development of the aptasensor were fitted using the Randles equivalent circuit as a model (see the inset of Figure 8B), and the simulated values of kinetic parameters were determined, and are marked along with the components of the equivalent circuit corresponding to this model (Figure 8B–F and Table 3). The components of the simple circuit with the following schematic representation—[**Rs**(C[**R_ct_** W])]—are the resistance of the solution (**R_s_**), which also includes the resistance of equipment and electrical circuit elements; the charge-transfer resistance (**R_ct_**); the double-layer capacitance (C); and the Warburg impedance (W). It was observed that the χ^2^ values were relatively small (Table 3), demonstrating that the data fit with Randle’s model [32,36]. The Nyquist plots of EIS obtained for bare C-SPE (Figure 8B; black) were fitted by using this simple equivalent circuit, while after the functionalization of the electrode with AuNPs (orange), the immobilization of the APT (blue), and the blocking step with MCH (green), the same equivalent circuit could be used, and C was replaced in the circuit by a constant phase element (CPE). This was because, in most practical situations, the EIS representations do not strictly follow the theoretically expected patterns, but show definite distortions, such as depressed semicircles that can be mathematically characterized by CPE [41,42]. A circuit in which a CPE is parallel linked with the W associated in series with **R_ct_** is characteristic of the porous structures or of the functionalized surfaces [41,43]. After the last step of the experimental protocol—namely, the immobilization of the analyte at the electrode—the equivalent circuit used for modeling the experimental EIS data was modified again, by introducing a series of R and C connected in parallel (see the inset in Figure 8F; red). This may be due to the change in the electron-transfer mechanism at the electrode surface, as well as the successful immobilization of 3-O-C_12_-HSL molecules [36].

#### 3.2.2. SPR Analysis

Characterization of the binding properties of APT is critical for studying APT molecular recognition. SPR is an optical detection method that provides a label-free and real-time characterization of the kinetic and steady-state affinity properties of biomolecular interactions. This technology has been used in the systematic development of ligands by exponential enrichment (SELEX) processes and in APT-based sensing applications. SPR sensors can register mass changes near the sensor surface that correlate with changes in the refractive index. These changes are recorded in real time, as sensorgrams in resonance units (RU) [44,45]. Typically, one interacting partner is immobilized on the chip surface, and its binding interactions with other molecules in the solution are monitored.

The interaction kinetics can be divided into three distinct phases: the association, steady-state, and dissociation phases of the molecules (Figure 9A) [33].

In our SPR setup, the specific APT was immobilized on a gold chip surface, and a series of 3-O-C_12_-HSL concentrations (10, 20, 50, 100, 200, 500, and 1000 nM) were injected at a constant flow rate. Based on these experiments, the binding parameters were analyzed and extracted.

The normalized, blank-subtracted sensorgrams were fit to a 1:1 kinetic binding model (Figure 9B) and a steady-state affinity model (Figure 9C). A ka of (8.97 ± 0.82) × 10^4^ M^−f^ s^−f^ and a *kd* of (7.76 ± 0.5) × 10^−3^ s^−s^ were determined. The *K_D_* of 87.1 ± 10 nM, calculated as the ratio of the binding rates (*K_D_,_kinetic_ = kd/ka*), compares well with that obtained from the steady-state affinity model (*K_D,equilibrium_*) (106.7 ± 9.3 nM). These values are close to the value reported in the literature [31]. This discrepancy could be caused by the use of different methods for the kinetic evaluation of APT–analyte binding, since Zhao et al. [31] determined the *K_D_* value based on a fluorescence assay using a microplate reader. Moreover, in our study, the APT was immobilized on a gold chip, while in the previous study, a 3-O-C_12_-HSL amino lactam surrogate was bound to streptavidin in each well on an ELISA plate, and the APT was then incubated at 37 °C for 45 min in each of these wells.

The APT showed fast-on and fast-off binding kinetics in the SPR experiments [44]. The changes in the SPR response during the association/dissociation phases support the binding capabilities of the APT for 3-O-C_12_-HSL. Moreover, the study showed that these changes increased proportionally to the concentration of the molecule.

### 3.3. Calibration Curve and Limit of Detection

To evaluate the analytical performance of the aptasensor, the optimized aptasensor was incubated with standard solutions of 3-O-C_12_-HSL at different concentrations (500 nM, 750 nM, 1 µM, 2.5 µM, 5 µM, 10 µM, 20 µM, 30 µM) under optimal conditions (room temperature, 30 min). The signal represented the percentage change in **R_ct_** obtained by EIS analysis before and after incubation with 3-O-C_12_-HSL, as described in Section 2.3.3. As shown in Figure 10A, the signal increased with the concentration of 3-O-C_12_-HSL; from 500 nM to 5 µM, the signal steeply increased, while for the 10–30 µM range the signal increase was moderate. Therefore, there is a good linear relationship between the signal and the logarithm of 3-O-C_12_-HSL concentration in the 500 nM–30 µM range (Figure 10B). The linear regression equation was calculated as y (signal) = 9.5429x (ln(conc.(µM))) + 31.1985, with a correlation coefficient (R^2^) of 0.9902.

The LOD was determined using Equation (4):**S_t_ ≥ S_b_ + 3s**(4)
where **S_t_** is the analyte signal, **S_b_** is the blank signal, and **s** is the standard deviation of five blank determinations. We found a limit of detection of 145 ng mL^−1^.

Table 4 presents the comparison of the analytical performance of the developed aptasensor with several other analytical methods. The aptasensor developed in this study presents the broadest dynamic range and an LOD that is low enough to allow the detection of 3-O-C_12_-HSL in different biological samples of practical relevance. Even though the aptasensor described in this study has a higher LOD than the ones obtained using the other methods, it exhibits several advantages compared to the other methods, including easy and low-cost fabrication, a simple sensing process, and providing a proof-of-concept of an aptasensor for 3-O-C_12_-HSL detection in real biological and bacteriological samples.

### 3.4. Interference Study

The selectivity of the aptasensor was tested by measuring the EIS response to two antibiotics (GEN and CFZ) and an antipyretic drug (APAP) used in the treatment of *P. aeruginosa* infections [48]. The selectivity towards other AI molecules structurally related to 3-O-C_12_-HSL, such as C_4_-HSL, C_12_-HSL, and 3-O-C_10_-HSL, was also evaluated. All of the molecules tested as interferents could be found together with 3-O-C_12_-HSL in the real samples. The EIS signal was recorded after incubation in 10 µM solutions containing the interferent alone and a mixture of the 3-O-C_12_-HSL and each interferent, and the response was compared with the signal corresponding to a 10 µM solution of 3-O-C_12_-HSL. All of the measurements were carried out in duplicate, and the signals in Figure 11 correspond to the average of the signals and the error bars to the relative standard deviations.

The aptasensor showed good selectivity towards 3-O-C_12_-HSL. This was due to the high affinity of the APT for 3-O-C_12_-HSL, allowing the APT to specifically recognize and capture the target molecule, leading to the change in its conformation.

The analysis of the three AHLs (C_4_-HSL, C_12_-HSL, and 3-O-C_10_-HSL) showed the importance of the lateral chain in the interaction with the APT, with the C_4_-HSL being bound by the APT in a smaller proportion than 3-O-C_12_-HSL, as was previously demonstrated by Zhao et al. [31], or leading to a smaller APT self-conformation changes. Even though the molecules similar to 3-O-C_12_-HSL (C_12_-HSL and 3-O-C_10_-HSL) are bound by the APT when alone in solution, the mixture with 3-O-C_12_-HSL still leads to good enough recoveries. The two antibiotics tested (GEN and CFZ) are large molecules, and tend to be adsorbed at the surface of the APT [49]; therefore, the analyses of the mixture of the antibiotic with 3-O-C_12_-HSL lead to a smaller signal because the access of 3-O-C_12_-HSL to immobilized APT is impaired.

The high affinity of the APT for 3-O-C_12_-HSL allows for obtaining good recoveries in the case of all analyte-interferent mixtures.

### 3.5. Analysis of Real Samples 

#### 3.5.1. Urine and Culture Media

The applicability of the developed aptasensor for the detection of 3-O-C_12_-HSL from real samples was tested by analyzing spiked human urine samples and spiked culture media, represented by NB. The samples were spiked and then treated with ethanol as described in Section 2.3.6, reaching a final concentration of 10 μM in the analyte. The final solution was incubated on the aptasensor using the optimal parameters. The concentration of 3-O-C_12_-HSL in the tested samples was determined using the calibration curve. This simple sample treatment and high selectivity of the aptasensor allowed us to obtain good recoveries for both samples, facilitating the detection of *P. aeruginosa* (Table 5). Urine did not seem to interfere with the analyte signal, and the NB had a higher signal, which can be given by one of its components that can bind to the APT, leading to further spatial rearrangement and, therefore, the increase in electron transfer.

#### 3.5.2. *P. aeruginosa* Cultures Analysis

The sensor was used to determine the concentration of 3-O-C_12_-HSL from two *P. aeruginosa* cultures (a standard strain (ATCC 27853) and a clinical isolate), at different moments in their growth (after 16 h, 24 h, 48 h, and 72 h). In parallel, the number of colonies (CFU mL^−1^) was estimated (Table 6).

The number of colonies grew exponentially over time, as expected (Figure 12A, black), and it could be observed that the concentration of 3-O-C_12_-HSL also increased exponentially (Figure 12A, red). This fact is consistent with data presented in the literature, shown that the amount of 3-O-C_12_-HSL found in the extracellular medium is proportional to the population density [3]. Correlating the lg of the concentration of 3-O-C_12_-HSL with the lg of the number of colonies, a linear fit was obtained, with Equation (5):**lg (No. of colonies) (CFU mL^−1^) = 2.049 × lg (conc. 3-O-C_12_-HSL (µM)) + 10.716**(5)
where R^2^ = 0.978 (Figure 12B). This shows a possible future application for the developed aptasensor, for the evaluation of the severity of infection with *P. aeruginosa*, correlating the concentration of 3-O-C_12_-HSL obtained with the bacterial growth.

## 4. Conclusions

A novel electrochemical aptasensor was developed for the selective, sensitive, and label-free detection of 3-O-C_12_-HSL—a molecule involved in the QS of *P. aeruginosa*, using the changes in the EIS spectra before and after incubation with the analyte. The use of AuNPs proved to be more efficient for the APT immobilization compared to the Au-SPE, so the C-SPE modified with AuNPs was used as the electrochemical platform for the immobilization of the thiol-modified aptamer. Each step for the elaboration of the aptasensor was characterized by CV, DPV, and EIS, using different redox probes (K_4_[Fe(CN)_6_], K_3_[Fe(CN)_6_], MB, Ru(NH_3_)_6_Cl_3_, and FcDM), the best results being obtained with the [Fe(CN)_6_]^4−^/[Fe(CN)_6_]^3−^ couple. A *K_D_* of 106.7 nM for the immobilized aptamer was determined by SPR. The developed aptasensor was able to successfully detect 3-O-C_12_-HSL in the 0.5–30 µM range, with an LOD of 145 ng mL^−1^ (0.5 µM). The aptamer-based electrochemical sensor showed good selectivity towards 3-O-C_12_-HSL detection, with limited influence from other AHLs, antibiotics, and paracetamol. The aptasensor was successfully used for the analysis of real samples (e.g., spiked urine samples, spiked microbiological growth media), with good recovery, and it could detect the variation in 3-O-C_12_-HSL concentration in microbiological cultures as the number of bacteria cells varied.

The described electrochemical aptasensor is the first example of QS molecule detection using a portable, accessible sensor, paving the way for rapid and easy identification and characterization of *P. aeruginosa* infections.

## Figures and Tables

**Figure 1 biosensors-12-00440-f001:**
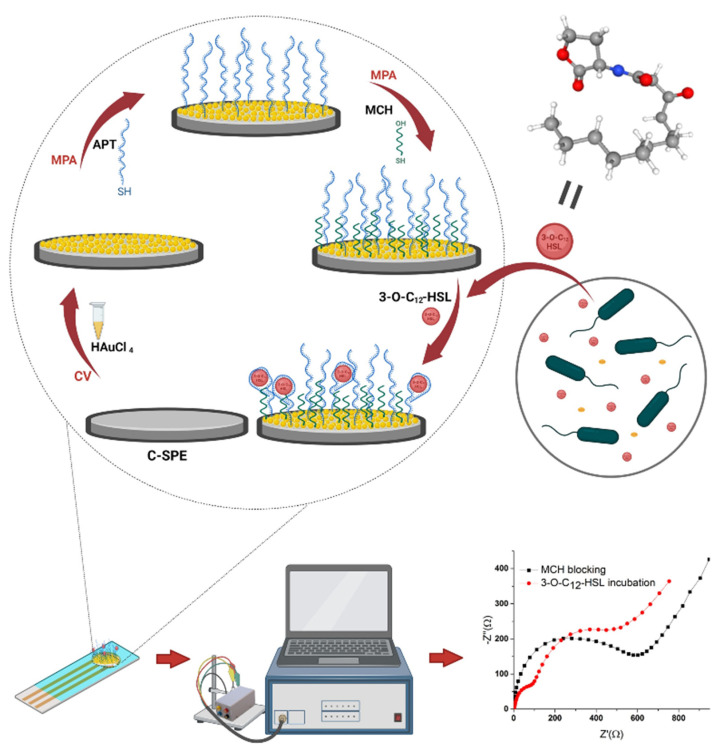
The schematic representation of the proposed aptasensor. Created with BioRender.com (accessed on 11 May 2022).

**Figure 2 biosensors-12-00440-f002:**
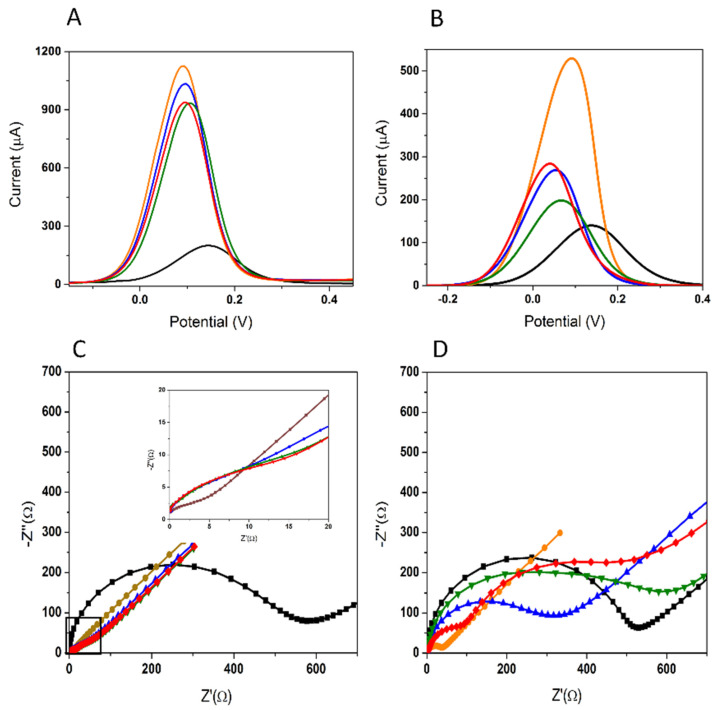
DPV and EIS analysis of 5 mM [Fe(CN)_6_]^3−/4−^ on the Au-SPE platform (**A**,**C**, respectively) and the C-SPE/AuNPs (**B**,**D**, respectively); bare electrode (**black**); electrode activation in H_2_SO_4_ (**A**,**C**; **brown**); AuNPs deposition from 1.5 mM HAuCl_4_ (**B**,**D**; **orange**); 1 µM APT immobilization by MPA (**blue**); 100 µM MCH blocking by MPA (**green**); incubation with 50 µM 3-O-C_12_-HSL for 30 min (**red**). Experimental conditions: DPV with a **SR** of 0.01 V s^−1^, a **PH** of 0.1 V, and a **PW** of 25 ms; and EIS with an **FR** from 0.01 to 100,000 Hz for a total of 50 frequencies, at open-circuit potential values.

**Figure 3 biosensors-12-00440-f003:**
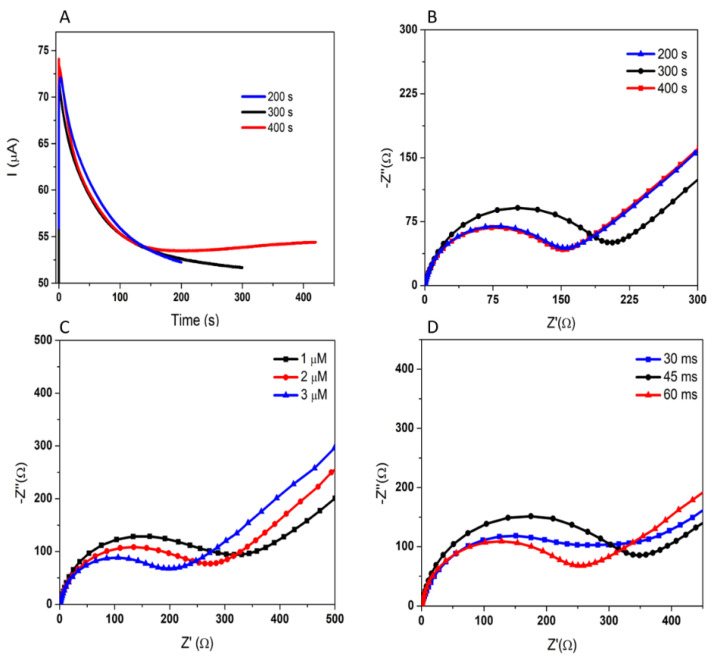
Optimization of the APT immobilization; the optimal parameters are represented in **black:** variation in the current during the APT immobilization by MPA (**A**); the Nyquist plots characterizing the APT immobilization after different durations (**B**); after different APT solution concentrations (**C**), and after different pulse durations (**D**).

**Figure 4 biosensors-12-00440-f004:**
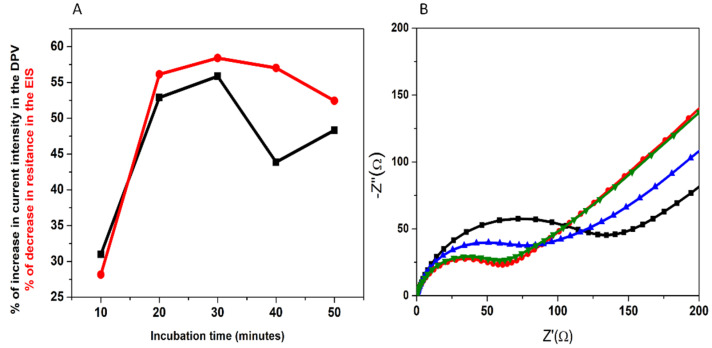
The influence of the incubation time on the electrochemical signal in DPV and EIS (**A**). The influence of the incubation dynamics (**B**): APT immobilization for static incubation (**black**); static incubation with 10 μM 3-O-C_12_-HSL (**red**); APT immobilization for dynamic incubation (**blue**); dynamic incubation with 10 μM 3-O-C_12_-HSL (**green**).

**Figure 5 biosensors-12-00440-f005:**
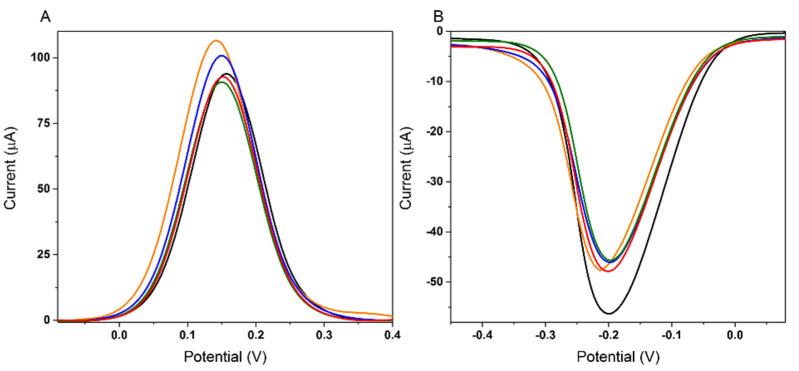
Electrochemical DPV characterization of the electrode surface after each modification step using the following as a redox probe: 1 mM FcDM in 0.1 M KCl (**A**); 5 mM Ru(NH_3_)_6_Cl_3_ in 0.1 M KCl (**B**) on bare electrode (**black**); C-SPE/AuNPs (**orange**); C-SPE/AuNPs/APT (**blue**); C-SPE/AuNPs/APT/MCH (**green**); and after incubation with 1 μM 3-O-C_12_-HSL (**red**).

**Figure 6 biosensors-12-00440-f006:**
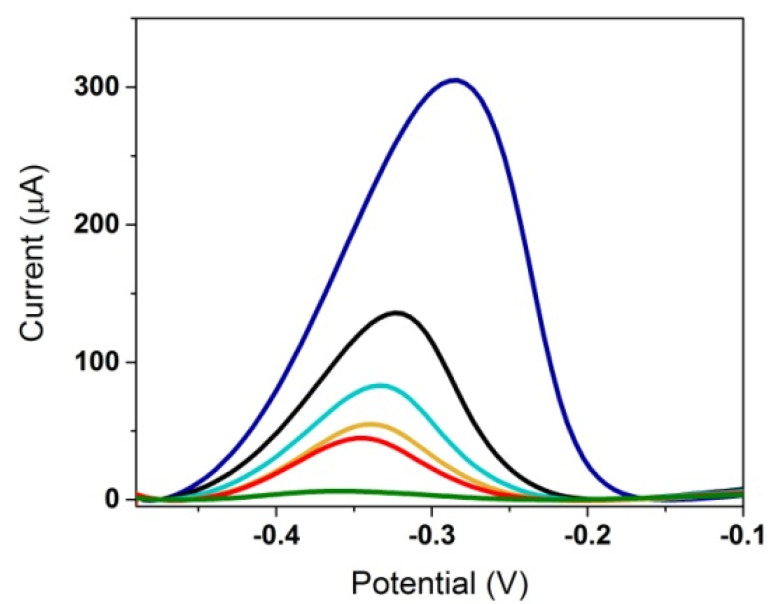
DPV characterization in 10 mM PBS pH 7.4 after 30 min incubation with 0.5 mM MB of the C-SPE/AuNPs/MCH (**green**); after 30 min incubation with 0.5 mM MB of the C-SPE/AuNPs/APT/MCH (**blue**); after 30 min incubation of the MB-labeled aptasensor with blank (**black**); 1 μM 3-O-C_12_-HSL (**cyan**); 10 μM 3-O-C_12_-HSL (**orange**); or 50 μM 3-O-C_12_-HSL (**red**).

**Figure 7 biosensors-12-00440-f007:**
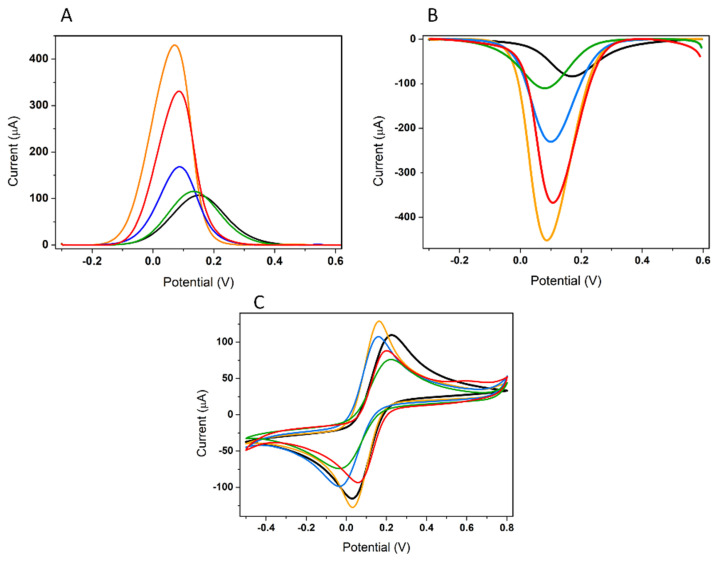
Electrochemical characterization of the electrode surface by DPV (oxidation) (**A**), DPV (reduction) (**B**), and CV after each modification step in 5 mM [Fe(CN)_6_]^3−/4−^ (**C**): bare electrode **(black)**; C-SPE/AuNPs (**orange**); C-SPE/AuNPs/APT (**blue**); C-SPE/AuNPs/APT/MCH (**green**); and after incubation with 50 μM 3-O-C_12_-HSL, for 30 min (**red**). Experimental conditions for DPV: SR of 0.01 V s^−1^, PH of 0.1 V, and a PW of 25 ms; and for CV: SR of 0.05 V s^−1^ and SP of 0.00244 V.

**Figure 8 biosensors-12-00440-f008:**
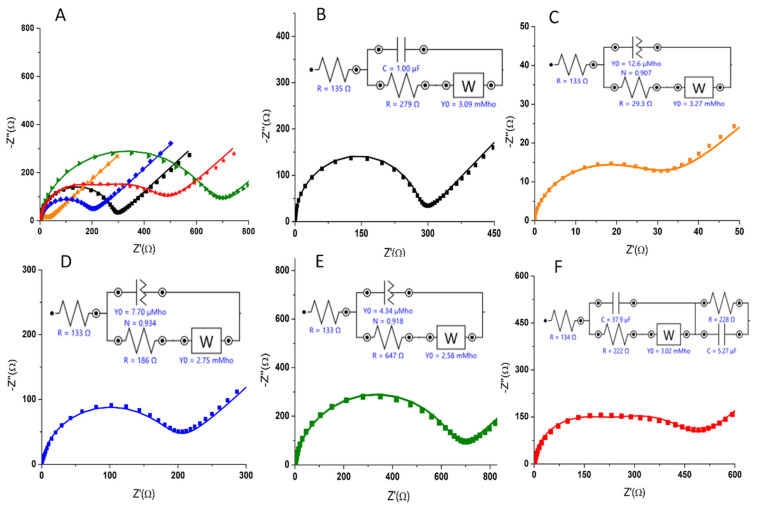
Nyquist plots and (**A**) the equivalent circuits used to fit (**B**) the impedance data of the bare electrode (**black**); (**C**) C-SPE/AuNPs (**orange**); (**D**) C-SPE/AuNPs/APT (**blue**); (**E**) C-SPE/AuNPs/APT/MCH (**green**); and (**F**) after incubation with 50 μM 3-O-C_12_-HSL for 30 min (**red**).

**Figure 9 biosensors-12-00440-f009:**
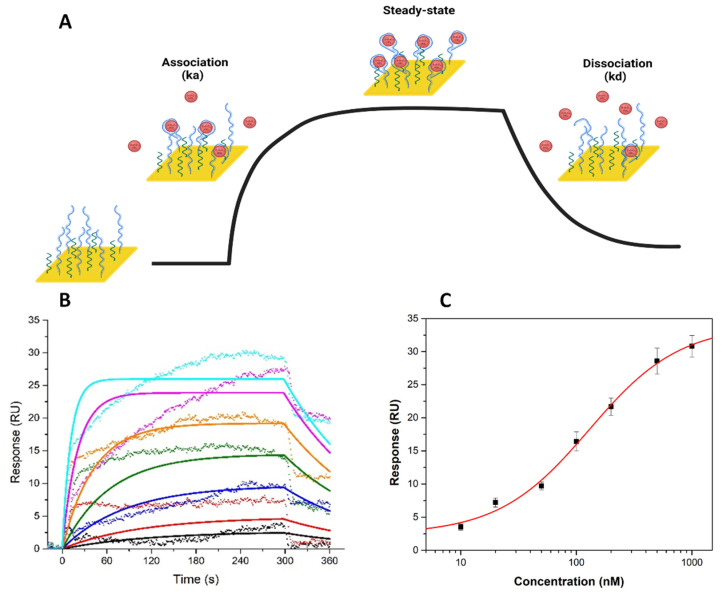
SPR characterization of the interactions between 3-O-C_12_-HSL and the selected APT: Schematic representation of the SPR-based APT binding assay. Created with BioRender.com (**A**). The kinetic analysis of 3-O-C_12_-HSL-APT binding behavior; representative normalized, blank-subtracted SPR sensorgrams for seven serial concentrations of 3-O-C_12_-HSL (**10 nM**, **20 nM**, **50 nM**, **100 nM**, **200 nM**, **500 nM,** and **1000 nM**) (**B**); the experimental points obtained (scatter) and the fitted curves (straight lines) (corresponding to a 1:1 kinetic binding model), and The corresponding equilibrium binding curve (fitted to a steady-state affinity model) (*n = 3*) (**C**). Experimental conditions: 60 µL/min flow rate, injections of 300 s, 60 s dissociation time.

**Figure 10 biosensors-12-00440-f010:**
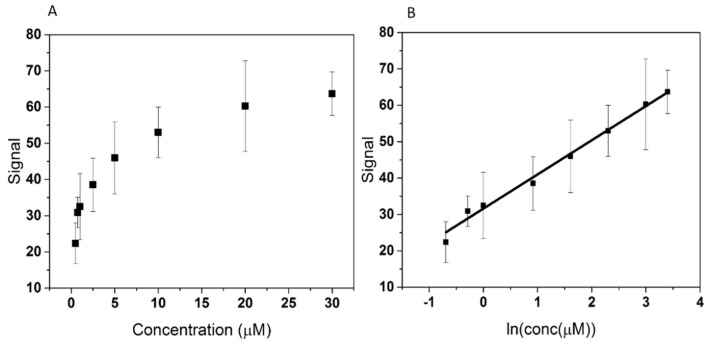
The correlation between the signal and the concentration of 3-O-C_12_-HSL (**A**), and linear fitting of the correlation between the signal and ln of the concentration of 3-O-C_12_-HSL (**B**).

**Figure 11 biosensors-12-00440-f011:**
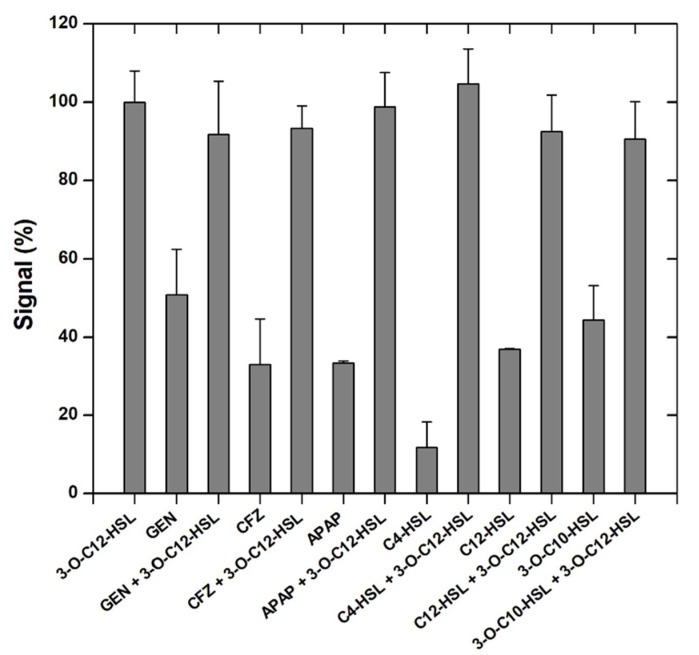
Aptasensor selectivity studies.

**Figure 12 biosensors-12-00440-f012:**
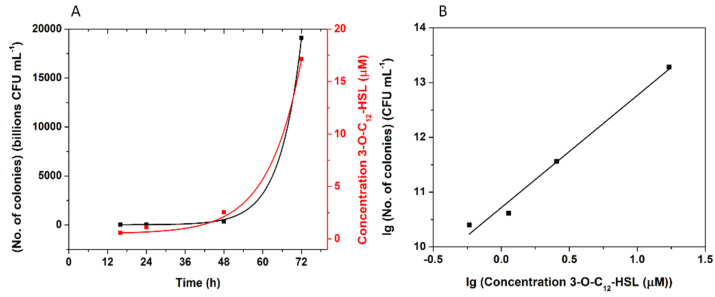
Correlation between the concentrations of 3-O-C_12_-HSL, determined with the sensors and the number of bacterial cells of *P. aeruginosa* ATCC 27853: The number of bacterial cells (**black**, left O–Y axis) and the concentration of 3-O-C_12_-HSL determined with the sensor (**red**, right O–Y axis) as a function of time (h) (**A**), and (**B**) linear fitting of the correlation between the lg of the number of bacterial cells and the lg of the concentration of 3-O-C_12_-HSL.

**Table 1 biosensors-12-00440-t001:** The performance of an Au-SPE surface vs. C-SPE/AuNPs with respect to the aptasensor development steps. ΔSignal (%) compared to the previous step.

Platform	Au-SPE		C-SPE/AuNPs	
	DPV (%)	EIS (%)	DPV (%)	EIS (%)
APT immobilization	−8.40	+207.18	−49.11	+532.8
MCH blocking	−9.58	+51.85	−26.14	+55.50
Incubation with 50 μM 3-O-C_12_-HSL	+0.21	−13.28	+42.92	−68.32

**Table 2 biosensors-12-00440-t002:** Signal modification in EIS and DPV after each optimized parameter.

Aptasensor Elaboration Step	Optimized Parameters		Signal Modification in EIS (%)	Signal Modification in DPV (%)
AuNP generation	HAuCl_4_ Conc. (μM)	1.5	−91	+205.68
		3	−88.42	+190.61
		5	−83.90	+168.38
	Number of CV cycles (Conc. HAuCl_4_ = 1.5 μM)	25	−82	+95.03
		35	−91	+197.64
APT immobilization	Immobilization technique	Overnight	-	−33.88
		MPA	-	−52.68
	APT solution concentration (μM) (MPA)	1	+407.06	−31.33
		2	+366.50	−22.08
		3	+341.62	−27.17
	Total immobilization time (s) (Conc. APT 1 µM)	200	+316.27	−23.10
		300	+433.26	−36.82
		400	+295.83	−23.62
	Pulse duration (s) (Conc. APT 1 µM, total immobilization time 300 s)	0.030	+616.30	−29.02
		0.045	+885.75	−43.06
		0.060	+565.47	−14.97
Incubation with 3-O-C_12_-HSL	Incubation time (min)	10	−3.41	+0.15
		20	−52.88	+56.12
		30	−55.85	+58.40
		40	−43.86	+57.04
		50	−48.32	+52.44
	Kinetic conditions (30 min)	Static	−52.41	+10.26
		Dynamic	−20.29	+6

**Table 3 biosensors-12-00440-t003:** Values of circuit elements obtained by fitting the experimental data obtained for each platform.

	R_S_ (Ω)	R_et_ (Ω)	W (mMho)	CPE (µMho)	N	C (µF)	χ2
**C-SPE**	135	279	3.09	-	-	1.00	0.003
**C-SPE/AuNPs**	133	27.20	3.25	-	-	5.79	0.001
**C-SPE/AuNPs/APT**	133	186	2.75	4.52	0.93	-	0.003
**C-SPE/AuNPs/APT/MCH**	133	647	2.58	4.34	0.91	-	0.006
**C-SPE/AuNPs/APT/MCH** **/** **3-O-C_12_-HSL**	134	222; 228	3.02	-	-	37.90; 5.27	0.01

**R_s_**—solution resistance; **R_et_**—resistance to charge transfer; W—Warburg element impedance; CPE—constant phase element; C—calculated capacitance; χ^2^—goodness of fit.

**Table 4 biosensors-12-00440-t004:** Comparison of this aptasensor with other methods previously reported for 3-O-C_12_-HSL detection.

Method	Linear Range (nM)	LOD (nM)	Sample	Ref.
HPLC-MS/MS	2.6–350	2.6	Culture media	[11]
Whole-cell biosensor (PA14-R3)	0.152–12	0.01	Culture media and CF sputum	[16]
Genetically engineered (electrochemical) biosensor	0.01–10	0.002	Liquid cultures and artificial saliva	[18]
Magnetic MIP-based electrochemical sensor	2.5–100	0.8	Bacteria supernatant samples	[27]
LasRV cell-free biosensor	5–100	4.9	CF sputum	[46]
Photoluminescence-based assay using cysteamine-capped TiO_2_ nanoparticles	10–160	10	Artificial urine	[47]
AuNPs-based electrochemical aptasensor	500–30,000	500	Spiked culture media, urine samples, culture media with bacteria	This work

**Table 5 biosensors-12-00440-t005:** Real analysis of samples spiked with 10 μM 3-O-C_12_-HSL.

Sample	Signal (%)	Spiked Conc. of 3-O-C_12_-HSL (μM)	Found Conc. of 3-O-C_12_-HSL (μM)	Recovery (%)
**Urine**	53.18	10	10.01	100.1 ± 8.59
**NB**	55.05	10	12.06	120.61 ± 7.84

**Table 6 biosensors-12-00440-t006:** *P. aeruginosa* cultures analysis.

Time	*P. aeruginosa* ATCC 27853		*P. aeruginosa* Clinical Isolate	
	No. of Colonies* (*CFU mL^−1^*)*	Conc. of 3-O-C_12_-HSL (µM)	No. of Colonies (CFU mL^−1^*)*	Conc. of 3-O-C_12_-HSL (µM)
**16 h**	2507 × 10^7^	0.582	1522 × 10^7^	0.383
**24 h**	4097 × 10^7^	1.131	2935 × 10^7^	0.675
**48 h**	36,455 × 10^7^	2.55	28,385 × 10^7^	3.06
**72 h**	1911 × 10^10^	17.14	1386 × 10^10^	7.76

## Data Availability

Not applicable.

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
