# Peer review of "Label-Free Electrochemical Aptasensor for the Detection of the 3-O-C12-HSL Quorum-Sensing Molecule in Pseudomonas aeruginosa"

_biosensors, 2022, doi:10.3390/bios12070440_

Round 1

Reviewer 1 Report

The manuscript entitled “Label-free electrochemical aptasensor for the detection of 3-O-C12-HSL quorum sensing molecule in Pseudomonas aeruginosa” by D.E. Capatina et al. reports the electrochemical sensor for the detection of 3-O-C12-HSL in Pseudomonas aeruginosa. The author did a systematic work, which contains rich data. However, some important evidences are necessary to support the conclusion The following points should be carefully addressed before publication.

1. In section 3.1.2, why the semicircle in EIS of the optimum parameters always is higher than those of other samples? As shown in the graph, the semicircle of samples (300 s, 1 mM, 45 ms) represented high resistance. Please further explain in detail.

2. The authors mentioned that “The APT immobilization was based on the interaction between the thiol function attached at the 3’ end of the APT and the AuNPs deposited on the carbon surface of the electrode when a S-Au covalent bond is formed.”. Please provide further results to confirm the interaction between them.

3. “The last step of the experimental protocol is represented by the binding of the target analyte to APT through the specific interaction, which determines the proximity of the two peaks at about 132 mV and the increase of their intensity (red).” What is the interaction between the APT and analyte (3-O-C12-HSL)? Please explain in more detail.

4. Please show the stability of aptasensor.

Author Response

Reviewer 1

The manuscript entitled “Label-free electrochemical aptasensor for the detection of 3-O-C12-HSL quorum sensing molecule in Pseudomonas aeruginosa” by D.E. Capatina et al. reports the electrochemical sensor for the detection of 3-O-C12-HSL in Pseudomonas aeruginosa. The author did a systematic work, which contains rich data. However, some important evidences are necessary to support the conclusion. The following points should be carefully addressed before publication.

We thank the reviewer for the suggestions which helps us improve the quality of the manuscript.

  1. In section 3.1.2, why the semicircle in EIS of the optimum parameters always is higher than those of other samples? As shown in the graph, the semicircle of samples (300 s, 1 mM, 45 ms) represented high resistance. Please further explain in detail.

We thank to the reviewer for this observation. First, the choice of the type of electrode used as the starting platform for the development of the aptasensor based on the voltammetry and electrochemical impedance data was made based on the largest variation of the signal recorded after the immobilization step of the aptamer compared to the previous step. Given that the immobilization of the aptamer caused an increase in the value of resistance to charge transfer, a higher increase of this parameter recorded on the C-SPE modified with AuNPs compared to Au-SPE was assimilated with the immobilization of a larger number of aptamer units, which leads to a better sensitivity for the detection of the analyte of interest. Subsequently, the optimization of the aptamer immobilization on the surface of the selected electrode was done by selecting the experimental conditions that allowed obtaining the largest and the most reproducible increase in the value of charge transfer resistance.

  1. The authors mentioned that “The APT immobilization was based on the interaction between the thiol function attached at the 3’ end of the APT and the AuNPs deposited on the carbon surface of the electrode when a S-Au covalent bond is formed.”. Please provide further results to confirm the interaction between them.

More information was added to text that supports the immobilization of the aptamer through S-Au covalent bonds (see paragraphs bellow).

The APT immobilization was based on the interaction between the thiol function attached at the 3’ end of the APT and the AuNPs deposited on the carbon surface of the electrode when a S-Au covalent bond is formed. The functionalization of AuNP via simple thiolate chemistry is a widely used method in many research areas and Au-S bond formation has been an intensively studied topic for decades due to its applications in sensor development [38]. The formation of the covalent Au-S bond is a complex process involving two stages: physisorption with the initial formation of a coordinative bond between the - SH group and Au while the H atom remains advantageously attached to the S atom, followed by chemisorption involving the dissociation of the S-H bond and the formation of the covalent Au-S bond due to the deprotonation of thiols and the formation of thiol radicals. The dissociated hydrogen atoms can be adsorbed on the gold surface or released as H2. The pH of the medium can affect the formation of the Au-S bond. A neutral or alkaline medium favors the dissociation of the S-H bond and the formation of the covalent bond [39]. The technique and parameters of this modification step are important as they influence the morphology of the APT layer and therefore the performance of the aptasensor. Two modifications techniques have been used: overnight incubation and MPA from a 1 μM APT solution. The potential assisted method showed better results with a signal decrease of 52.68 % compared to the overnight incubation, which determined a decrease in current intensity of 33.88 % (Table 2). These results show that the MPA method promotes a better immobilization of the APT and an enhanced formation of Au-S bonds between the APT and AuNP, leading to the assembly of a more organized and stable layer. The Au-S interaction and the efficiency of the immobilization method employed in this study were also reported by Ge et al. [40] and Jambrec et al. [41]. When shifting the potential of zero charge between two values, the DNA layer assembles with rapidity and offers the possibility to obtain reproducible surfaces, which is essential for analytical performance.

  1. “The last step of the experimental protocol is represented by the binding of the target analyte to APT through the specific interaction, which determines the proximity of the two peaks at about 132 mV and the increase of their intensity (red).” What is the interaction between the APT and analyte (3-O-C12-HSL)? Please explain in more detail.

As mentioned in the manuscript (lines 638-644), the SELEX selection of the aptamers was performed by Zhao et al. [27], where fluorescence affinity studies were performed to determine the most selective ssDNA sequences towards 3-O-C12-HSL. KD values between 20 nM and 35 nM were obtained for 3 aptamers and we have selected the one with the lowest dissociation constant.

Hence, the characterization of the aptamer affinity was studied in this work by SPR analysis. The results were discussed in lines 574-588. As mentioned, the SPR measurements were performed in the same conditions to mimic the electrochemical assay and better determine the affinity towards 3-O-C12-HSL.

  1. Please show the stability of aptasensor.

Due to the advantages of the electrochemical modification of the electrode, the fabrication of the aptasensor does not require a lot of time, so the aptasensors were tested in the same day they were fabricated. Since we use SPE, the aptasensor is disposable, single use, there is no need to test the stability of the aptasensor for several analysis.

Reviewer 2 Report

Journal: Biosensors

Article title: Label-free electrochemical aptasensor for the detection of 3-O-12-HSL quorum sensing molecule in Pseudomonas aeruginosa

The authors have developed the N-3-oxo-dodecanoyl L-homoserine lactone (3-O-C12-HSL) sensors using carbon-based SPE with a detection limit of 0.5 µM. it is an interesting research article and its scope match the journal.

I recommend its publication with minor revision.

1.                   The graphical abstract is necessary

2.                   In the introduction the authors need to mention the aptamers applications in electrochemical sensing in a detailed way.

3.                   Corresponding Bode plots of Fig 8 have to be mentioned

4.                   How authors are authenticating the reliability of the equivalent circuits of the respective impedance plots

5.                   What is the reason for huge errors  (Fig 10 and Fig 11)

6.                   Why the authors are not doing any surface characterization of carbon-SPE (like FESEM-EDX or TEM)

7.                   Few reference numbers are missing line numbers 50, 58, and 73 authors need to correct them.

8.                   The equation 2 has to be given with proper citation

9.                   Can AFM results further correlate with STM results authors need to discuss it

10.               Conclusion has to be improved with deep insights (it looks very general)

11.               I suggest the authors make a general table and discuss the advantages and disadvantages of labe and label-free electrochemical aptasensors.

12.                Please check the grammatical and syntax error

13.               There has to be a graphical abstract that can create curiosity for the readers and present the work aesthetically.

Author Response

Reviewer 2

The authors have developed the N-3-oxo-dodecanoyl L-homoserine lactone (3-O-C12-HSL) sensors using carbon-based SPE with a detection limit of 0.5 µM. it is an interesting research article and its scope match the journal.

I recommend its publication with minor revision.

We thank the reviewer for the suggestions which helps us improve the quality of the manuscript.

  1. The graphical abstract is necessary

We thank the reviewer for the suggestion. A graphical abstract was added.

  1. In the introduction the authors need to mention the aptamers applications in electrochemical sensing in a detailed way.

The introduction has been improved integrating the reviewer’s suggestion. The following paragraphs were added:

The combination of electrochemical techniques with APTs has promoted the development of aptasensors with many applications in electrochemical sensing for a wide range of molecules (drugs, biomolecules, toxins, pesticides), ions, and pathogens [3,27–29]. The electrochemical aptasensing device usually consists of the APT-based sensing layer, the transducer that converts the resulting chemical signal into an electrical output, and the analyzer that has a readout function [28]. Depending on the electrochemical techniques used and the signal measured (impedance, current and potential), aptasensors can be impedimetric, amperometric/voltammetric or potentiometric [27,29]. Various electrochemical aptasensors have been developed based on different measurement techniques such as differential pulse voltammetry (DPV), cyclic voltammetry (CV), electrochemical impedance spectroscopy (EIS), and field effect transistors (FET). Electrochemical aptasensors can be divided into labeled (with enzymes, antibodies, metal nanoparticles, or redox compounds) or label-free aptasensors, depending on the detection methods used. The labels can be covalently or non-covalently attached to APTs. In both cases, the presence of targets is indicated by changes in either current or capacitance, and the concentration of the analyte can be directly correlated with the change in the electrochemical feature [28,29]. Label-free aptasensors are favored to the detriment of label-based sensors because of their advantages, such as less complicated design, reduced preparation time, analytical quality, wider dynamic range, relatively fast reaction time, easy operation, and cost-effectiveness by eliminating the need for complex labels [30,31]. It is important to note that label-free electrochemical aptasensors, in addition to their noticeable advantages, have some disadvantages, such as lower selectivity than similar labeled electrochemical aptasensors and a low signal-to-noise ratio at low target concentrations, which requires mediator composites to amplify the output signal [30].

Overall, the electrochemical aptasensors have several advantages, such as high sensitivity and specificity, low cost, ease of use, and the possibility of miniaturization, making them useful tools for biomedical, food, or environmental analysis in the field. However, one challenge in developing an aptasensor for a particular target is the availability of the specific APT for that analyte.

  1. Corresponding Bode plots of Fig 8 have to be mentioned

We thank to the reviewer for this observation. We have chosen to present in the manuscript the Nyquist plots of the EIS data because they are more common, are more often used in publications and more accessible to the reader less familiar with this electrochemical technique. Of course, the software allows us to represent the Bode plots of EIS, and we have included these representations here. In our opinion, the data already included in the manuscript, namely the Nyquist representations, are sufficient to demonstrate the successful elaboration of the aptasensor. If the reviewer believes that the information provided by the Bode charts is indispensable, we may consider including it in the Supplementary data.

Bode representation for: bare electrode C-SPE (black points; left) and after AuNPs deposition from 1.5 mM HAuCl4 (orange points; right).

Bode representation for C-SPE/AuNPs after APT immobilization (1 µM) by MPA (blue points; left) and for C-SPE/AuNPs/APT after MCH blocking by MPA (100 µM) (green points; right)

Bode representation for: C-SPE/AuNPs/APT/MCH after incubation with 50 µM 3-O-C12-HSL for 30 minutes (red poins).

  1. How authors are authenticating the reliability of the equivalent circuits of the respective impedance plots

We thank to the reviewer for this observation. The equivalent circuits used for fitting and simulations of the experimental EIS data were selected taking into account the experience of the authors in the field of electrochemical characterization of simple electrodes and after the functionalization with nanomaterials and biomimetic elements such as aptamers. The selected equivalent circuits were then validated by comparison with literature data. Thus, in the manuscript 4 bibliographic sources are cited meant to justify each modification of the components of the selected equivalent circuit.

According to literature study, the simple Randles equivalent circuit having the following schematic representation, [Rs(C[Rct W])], was selected for the bare C-SPEs, while after the functionalization of the electrode with AuNPs, the immobilization of the aptamer and the blocking step with MCH, the capacitance (C) element was replaced in the circuit by constant phase element (CPE), since a circuit in which a CPE is parallel linked with the diffusion element (W) associated in series with Rct is characteristic to the porous structures or to the functionalized surfaces. After the immobilization of the analyte at the electrode, the equivalent circuit used for modeling the experimental EIS data was modified again, by introducing a series of R and C connected in parallel because the successful immobilization of these molecules at the electrode caused the change in the electron transfer mechanism.

  1. What is the reason for huge errors  (Fig 10 and Fig 11)

The big errors are because, at high concentrations of the analyte, different interaction processes occur (leading to changes in the Nyquist plots) at the surface of the aptasensor. The signal is based on the differences in the aptamer conformation before and after incubation with the analyte and this is associated with certain variability.

  1. Why the authors are not doing any surface characterization of carbon-SPE (like FESEM-EDX or TEM)

We appreciate the reviewer’s suggestion. The C-SPEs are commercially available, and their surface characterization is performed by the manufacturer. Moreover, the modification of the surface of the C-SPE was intensively studied in our previous papers, one of the recent papers being published by Blidar et al. (10.1016/j.foodchem.2021.131127). The study by Blidar et al. aimed for the characterization of screen-printed electrodes modified with different gold nanostructures, therefore, surface characterization was performed by SEM, AFM and XPS. However, the present study focused on the aptasensing approach, and the major interest was the electrochemical characterization of the steps involved in the sensor development. 

  1. Few reference numbers are missing line numbers 50, 58, and 73 authors need to correct them.

We thank the reviewer for the suggestion. Reference numbers were added in several paragraphs (marked in the manuscript).

  1. The equation 2 has to be given with proper citation

The equation 2 was created by us and it is cited in the previous paragraph.

  1. Can AFM results further correlate with STM results authors need to discuss it

As stated in the answer for the question 6, the novelty of this study relies on the aptasensing approach, being the first electrochemical aptasensor for the detection of 3-O-C12-HSL. The Materials & Methods section provides the details related to the deposition of Au at C-SPE by adapting one of our previous works [28] with the purpose of immobilizing the thiol-tethered APT and it was successfully characterized by CV, DPV and EIS to confirm the electrode modification steps. The AFM analysis would not bring new, relevant informations. 

  1. Conclusion has to be improved with deep insights (it looks very general)

The conclusions have been improved by adding more details.

  1. I suggest the authors make a general table and discuss the advantages and disadvantages of labe and label-free electrochemical aptasensors.

The suggested information was added in the introduction:

Label-free aptasensors are favored to the detriment of label-based sensors because of their advantages, such as less complicated design, reduced preparation time, analytical quality, wider dynamic range, relatively fast reaction time, easy operation, and cost-effectiveness by eliminating the need for complex labels [30,31]. It is important to note that label-free electrochemical aptasensors, in addition to their noticeable advantages, have some disadvantages, such as lower selectivity than similar labeled electrochemical aptasensors and a low signal-to-noise ratio at low target concentrations, which requires mediator composites to amplify the output signal [30].

  1.  Please check the grammatical and syntax error

We thank the reviewer for this remark. We corrected the grammatical and syntax errors using the pieces of advice provided by the journal.

  1. There has to be a graphical abstract that can create curiosity for the readers and present the work aesthetically.

We thank the reviewer for the suggestion. A graphical abstract was added.
